# Enhanced Efficacy against Drug-Resistant Tumors Enabled by Redox-Responsive Mesoporous-Silica-Nanoparticle-Supported Lipid Bilayers as Targeted Delivery Vehicles

**DOI:** 10.3390/ijms25105553

**Published:** 2024-05-20

**Authors:** Shuoye Yang, Beibei Zhang, Xiangguo Zhao, Mengwei Zhang, Mengna Zhang, Lan Cui, Lu Zhang

**Affiliations:** 1School of Biological Engineering, Henan University of Technology, Zhengzhou 450001, China; 13253432870@163.com (B.Z.); zxg147258@outlook.com (X.Z.); zmw15649858235@163.com (M.Z.); zmn2004@outlook.com (M.Z.); cuilanmm56@163.com (L.C.); zhanglu@haut.edu.cn (L.Z.); 2Key Laboratory of Functional Molecules for Biomedical Research, Zhengzhou 450001, China

**Keywords:** multidrug resistance, mesoporous silica nanoparticles, redox-responsive, antitumor effect, nano-delivery vehicles

## Abstract

Multidrug resistance (MDR) is frequently induced after long-term exposure to reduce the therapeutic effect of chemotherapeutic drugs, which is always associated with the overexpression of efflux proteins, such as P-glycoprotein (P-gp). Nano-delivery technology can be used as an efficient strategy to overcome tumor MDR. In this study, mesoporous silica nanoparticles (MSNs) were synthesized and linked with a disulfide bond and then coated with lipid bilayers. The functionalized shell/core delivery systems (HT-LMSNs-SS@DOX) were developed by loading drugs inside the pores of MSNs and conjugating with D-α-tocopherol polyethylene glycol 1000 succinate (TPGS) and hyaluronic acid (HA) on the outer lipid surface. HT-LMSNs-SS and other carriers were characterized and assessed in terms of various characteristics. HT-LMSNs-SS@DOX exhibited a dual pH/reduction responsive drug release. The results also showed that modified LMSNs had good dispersity, biocompatibility, and drug-loading capacity. In vitro experiment results demonstrated that HT-LMSNs-SS were internalized by cells and mainly by clathrin-mediated endocytosis, with higher uptake efficiency than other carriers. Furthermore, HT-LMSNs-SS@DOX could effectively inhibit the expression of P-gp, increase the apoptosis ratios of MCF-7/ADR cells, and arrest cell cycle at the G0/G1 phase, with enhanced ability to induce excessive reactive oxygen species (ROS) production in cells. In tumor-bearing model mice, HT-LMSNs-SS@DOX similarly exhibited the highest inhibition activity against tumor growth, with good biosafety, among all of the treatment groups. Therefore, the nano-delivery systems developed herein achieve enhanced efficacy towards resistant tumors through targeted delivery and redox-responsive drug release, with broad application prospects.

## 1. Introduction

Nowadays, chemotherapy is still the most effective and frequently used treatment approach for cancer therapy, even though it may cause some severe side effects [1]. However, the prolonged and efficient use of chemotherapeutic agents is often limited by drug resistance that occurs shortly after the treatment. More seriously, in many clinical cases, cancer evolves to be simultaneously resistant to a variety of chemotherapeutics with different structures and mechanisms, termed multidrug resistance (MDR) [2]. After the occurrence of MDR, the surviving cancer cells become insensitive or do not respond to drugs, resulting in tumor recurrence and progression. MDR is currently estimated to be the leading reason for the poor long-term therapeutic effect of chemotherapy. In total, MDR is responsible for over 90% of treatment failures in clinical applications [3].

The mechanism of tumor MDR is primarily based on the overexpression of drug efflux pumps in the cellular membrane. The prominent efflux pump is P-glycoprotein (P-gp), belonging to the ATP-binding cassette superfamily of proteins [4]. Over the past decades, various strategies, such as using small molecule inhibitors and gene therapy, have been exploited to overcome MDR. Among these strategies, nanomedicine formulations or nanoscale delivery systems (DDSs) can take advantage of the enhanced permeability and retention (EPR) effect or ligand-receptor-mediated active targeting to deliver agents specifically to cancer cells and enhance drug accumulation in tumor sites. Therefore, they are regarded to hold significant potential in improving therapeutic efficacy for resistant malignancies [5,6]. However, insufficient drug release from nanoparticles (NPs) during circulation and low drug concentration inside tumor cells still impede the effectiveness of nano-delivery systems [7].

In addition to ATP-dependent drug efflux, the tumor microenvironment (TME) also plays a key role in the rapid development of MDR. The TME is commonly regarded as a highly complex environment surrounding tumors. The heterogeneity inside the TME is extremely prone to leading to resistance to traditional chemotherapy strategies [8,9]. Generally, the TME is characterized by variable oxygen concentrations, low pH, high levels of reactive oxygen species (ROS) and glutathione (GSH), and so on [10]. Based on these features, a variety of TMEs or stimuli-triggered NPs have been developed for the delivery of anticancer agents to achieve responsive release in tumor sites [11]. The stimuli-responsive delivery systems provide a promising opportunity to trigger drug release in an on-demand manner and exhibit many superiorities over conventional nanomedicines for the reversal of MDR. For instance, the GSH concentration in drug-resistant tumor cells is up to approximately 1000 times higher than in the extracellular fluids [12]. Disulfide bonds (-SS-) are the most commonly used reducible linkers. They can be inserted into the nanomaterials to act as a linkage to bind ligands onto carriers or as “gatekeepers” to block cargo leakage. After cleavage of the disulfide bond via thiolysis in the presence of GSH, the structural transformation will rapidly lead nanocarriers to disintegrate to trigger intracellular drug release [13,14]. Thus, as smart carriers, redox-responsive delivery systems have attracted great attention for achieving site-specific accumulation and triggering drug release in response to physicochemical properties in pathological conditions [15,16].

As one of the novel delivery approaches, shell–core (SC) nanostructures are extensively applied for transporting chemotherapeutics, nucleic acids, or immunomodulators, which can significantly enhance their anticancer efficacy and reduce toxic effects [17,18]. SC NPs are generally characterized as a nanosystem with a hybrid structure comprising an exterior shell and an inner core. Compared to nanocarriers with a single function, SC NPs have the potential to facilitate a combination of materials with different features and functions [19,20]. Typically, the core materials in SC NPs can efficiently encapsulate therapeutic cargos, while the shell materials provide outer barriers to hinder unnecessary cargo leakage and ease surface modification with functional groups. Therefore, SC NPs are widely used for the design of multi-functional delivery systems with high drug loading capacity and bioavailability and efficient drug targeting and release [21,22]. Liposomes with a particle size range from 50 to 300 nm are representative lipid-based nanocarriers, and they are generally characterized as vesicles in which one lipid bilayer encloses an aqueous space. Other kinds of NPs with smaller sizes may be encapsulated into an aqueous core to form SC nanostructures. As one of the inorganic materials, mesoporous silica NPs (MSNs) have a series of appealing properties, including a uniform and tunable pore size, a large surface area and pore volume, ease of synthesis and drug loading, and good physiochemical stability. Moreover, the surface of MSNs can be physically coated and/or chemically conjugated using organic/inorganic moieties with different properties and functions, facilitating the design of SC-nanostructured carriers for the delivery of various therapeutic agents. MSN-supported lipid bilayers (LMSNs) are hybrid nanostructures formed through encapsulation of the MSN core within an outer lipid bilayer; they have been verified to possess several unique properties superior to most SC nanocarriers [23,24]. Hybrid LMSNs can synergistically combine the individual advantages of MSNs and liposomes. For instance, MSNs can act as a skeleton core to supply support to stabilize the outer lipid bilayer, while the lipid components have the potential to enhance the biocompatibility and affinity with the cell membrane. Meanwhile, the dual-encapsulation from MSNs and lipid layers enables LMSN nanocarriers to accommodate high payloads for disparate cargo [25]. D-α-tocopherol polyethylene glycol 1000 succinate (TPGS) is reported to possess the ability to inhibit drug efflux by suppressing the activity of P-gp ATPase in an ATP-dependent manner. Therefore, TPGS can act as an uncompetitive inhibitor of P-gp ATPase to block the energy supply for P-gp pumps, which favor drug internalization into tumor cells to overcome MDR. In addition, TPGS can boost the anticancer efficacy of drugs to realize synergistic effects by inducing the apoptosis of cancerous cells. Hyaluronic acid (HA) is involved in regulating a series of biological functions, including cell proliferation and differentiation. HA can also specifically bind to various cell-surface receptors, such as the CD44 cluster, which is generally overexpressed on the surface of tumor cells. Thus, it is extensively applied to construct active-targeting drug delivery vehicles in the pharmaceutical field. In addition to serving as the targeting ligand, the multiple functional groups available on the surface of HA endow HA-based nanocarriers with good biocompatibility and physiological stability [26,27].

Herein, a multi-functional, stimuli-responsive LMSN delivery system was developed to improve therapeutic efficacy against resistant tumors. MSNs were fabricated and then linked with an aminated disulfide bond that acts as a “pore blocker”. The outer lipid bilayers coated onto MSNs were conjugated with TPGS and HA. Doxorubicin (DOX) was used as a model drug to be encapsulated by functional LMSNs for the treatment of resistant human breast cancer (MCF-7/ADR). The in vitro drug release properties, cytotoxicity, cellular uptake efficiency and mechanism analysis, and inhibition of P-gp-mediated efflux were systematically investigated. Lastly, the in vitro and in vivo antitumor activity of DOX-loaded LMSN systems were evaluated in MCF-7/ADR cells and tumor-bearing nude mice, respectively.

## 2. Results and Discussion

### 2.1. Preparation and Characterization of Various MSN-Based Nanocarriers

The synthesis, preparation procedure, and antitumor mechanism of functionalized LMSNs are described in Figure 1. The MSN “cores” were manufactured using the classical templated self-assembly method, in which DEA acted as a base catalyst to tailor the particle sizes of nanoparticles. MSNs were then conjugated with amino and carboxyl groups successively to control the position of functionalization and prevent the premature “pore-blocking” effect caused by the addition of organ silane. Subsequently, the cleavable disulfide bonds (S-S) were covalently grafted onto MSNs using Cys·HCl as a donor to obtain MSNs-SS. The outer “shells” were further formed through the fusion of lipid vesicles on monosized MSNs-SS to produce bilayer-coated composite nanoparticles. Afterward, LMSN carriers were functionalized with HA and TPGS to improve their delivery properties, and the chemotherapeutic drug DOX was loaded through π-π conjugation inside the pores of MSNs [28].

From the TEM and SEM images in Figure 1, the MSNs showed a uniform and ordered morphology, and they had an average diameter lower than 100 nm, with a disc shape in the cross-section (Figure 1G,H). The three-dimensional mesoporous structure with a large number of pores on the surfaces of MSNs and MSNs-SS could also be clearly observed (Figure 1A,B). As for LMSN samples (LMSNs-SS, T-LMSNs-SS, and HT-LMSNs-SS), their surfaces were rough and appeared to be spherically shaped. The characteristic fingerprint-link ring around MSNs (thickness range of 3–7 nm, approximately) proved the presence of a lipid layer and a core–shell structure (Figure 1D–F). In addition, the thickness of the film-like ring in T-LMSNs-SS and HT-LMSNs-SS was larger than in LMSNs-SS.

The measurement results of the average particle size, zeta potential, and polydispersity index (PDI) are listed in Table 1 and shown in Figure 2A,B. The particle sizes of MSNs and MSNs-SS were about 85 nm and 123 nm, respectively, while the LMSN samples had larger sizes (higher than 180 nm), indicating a significant increase in the diameter of MSN NPs after lipid bilayer coating. It should be noted that the LMSN samples were apt to cluster after lipid coating, which accounts for the size increase. Although the PDI of liposomes and LMSNs-SS became larger than that of MSNs, the functionalized T-LMSNs-SS and HT-LMSNs-SS showed the lower PDI. Moreover, the MSNs were negatively charged, with a zeta potential value of −10.87 mV, which was caused by the silanol group (Si-OH) on their surface. After attachment with -SS-NH_2_, the potential value of MSNs-SS was dramatically increased to 35.13 mV due to the introduction of positively charged amino groups [29]. The zeta potential of LMSNs-SS was further decreased to 6.25 mV after lipid bilayer coating resulting from the shielding effect of the neutrally charged lipid components.

The adsorption–desorption isotherms and pore size distribution curves are shown in Figure 2C–F. The specific surface areas of MSNs and MSNs-SS were 512 m^2^/g and 23 m^2^/g, respectively, whereas after coating MSNs with a hybrid lipid layer, the surface area dropped sharply to lower than 0.3 m^2^/g. Although the pore volume of MSNs decreased from 2.58 nm to 2.38 nm after attachment with -SS-NH_2_, the pore channels were still sufficient for cargo loading with high efficiency. Nevertheless, the pore size distribution curves of LMSN samples were almost straight lines, indicating that the pores on the surfaces of MSNs were completely blocked to become undetectable. Thus, drug loading was performed prior to the lipid coating procedure. Similarly, the pores of drug-loaded MSNs were undetectable, which suggested that their mesoporous surface was occupied by DOX molecules, and MSNs had high drug-loading capacity.

The FT-IR spectra of various nanocarriers and nano-formulations are shown in Figure 3A,B. The stretching vibration peaks at 800–1500 cm^−1^ and 3500 cm^−1^ were the characteristic spectra of the Si-O-Si and silanol group [30], respectively, while the weak peak at 2900 cm^−1^ indicated the removal of template agent CTAB from MSNs. The characteristic absorption peak of the disulfide bond appearing at 1638 cm^−1^ proved the attachment of -SS-NH_2_ onto MSNs, which was significantly weakened after lipid bilayer coating for LMSN samples. The vibration peaks of liposomes were mainly detected at 1240 cm^−1^ and 1740 cm^−1^, denoting P=O and C=O stretching bands, respectively. For T-LMSNs-SS and HT-LMSNs-SS, the peaks around both 1278 cm^−1^ and 1342 cm^−1^ represented the intense absorbance of a methyl group (-CH_3_) in TPGS, and the P=O and C-H stretching vibrations from long-chain fatty acids were apparently enhanced. Moreover, after loading of DOX, the absorption peaks at 3325 cm^−1^ (indicating -OH), 2924 cm^−1^ (indicating -CH_2_-), 1730 cm^−1^ (indicating C=O), and 991 cm^−1^ (indicating C-H vibration peak) of each drug-loaded sample were weakened or disappeared, which suggested that most of DOX molecules were encapsulated inside the nanocarriers.

In Figure 3C, MSNs and MSNs-SS-NH_2_ had intense phase transition peaks at 60 °C and 75 °C, respectively. The endothermic peaks at about 112 °C detected for liposomes (freeze-dried samples) demonstrated the crystalline phase transition from a solid to a gel state. Compared with liposomes, the curve feature of LMSNs showed no marked change, although their phase transition temperature (Tm) decreased slightly. For T-LMSNs-SS and HT-LMSNs-SS, an endothermic peak appearing at about 112 °C may be attributed to their crystalline transition by melting after surface modification. In XPS Survey spectra (Figure 3D), the binding energy peak was detected at 103.7 eV for MSNs and MSNs-SS, which was the characteristic peak of the silicon element (Si2p). The intense S2p peak appearing at about 164 eV in the XPS curve of MSNs-SS further proved the presence of -SS-NH_2_ on the surface of MSNs. Both the Si2p and S2p peaks were observed to be sharply weakened in the XPS curve of liposome and LMSN samples, revealing that MSN particles were efficiently coated by a lipid bilayer to produce hybrid LMSNs. XRD measurement results (Figure 3E) showed that the diffraction peaks of MSNs, liposomes, and LMSNs-SS were detected at 22°, 19°, and 20°, respectively, revealing the amorphous state of these carriers. The diffraction patterns at 20° and 25° of functionalized LMSNs were weak and blunt, indicating that the stability of LMSN carriers would be enhanced to a certain extent after surface modification. In the Raman spectrum shown in Figure 3F, MSNs had the characteristic bands at 1450 cm^−1^ from -OH and at 2900 cm^−1^ from silanol groups, respectively, whereas the peak detected at 1450 cm^−1^ for liposomes was caused by -OH in phospholipid molecules. MSNs-SS also showed a characteristic stretching vibration at 700 cm^−1^. In comparison with MSNs-SS or liposomes, the peak intensity at 700 cm^−1^ and 1450 cm^−1^ was decreased for LMSN samples, demonstrating the tight coverage of the lipid layer onto MSN “cores”. Furthermore, the FT-IR and Raman signal from the lipid layer of HT-LMSNs-SS become notably weak, suggesting that the dual modification with FA and TPGS had significant effects on the surface chemical properties of LMSNs.

### 2.2. Drug-Loading Efficiency and In Vitro Drug Release

A rapid and accurate HPLC assay was developed for quantitative analysis of DOX. The retention time (RT) of DOX was about 5.4 min, whereas MSNs and lipid materials were eluted within 3 min. The calibration curve of CoQ10 is calculated as A = 10,469 × C − 15,429 (r = 0.9999), showing a good linear relationship from 0 to 200.0 µg/mL. The intra-day and inter-day precision (RSD) of DOX were 0.20% and 0.13%, respectively. The recovery of the QC samples was 95.92%, 99.03%, and 103.16%, with RSD of 0.83%, 0.75%, and 0.60% (*n* = 5). The above results were all within the acceptable range. The encapsulation efficiency (EE) and drug-loading efficiency (LE) listed in Table 2 showed that all nano-formulations possessed a high drug-loading capacity [31]. For instance, the LEs of MSNs and MSNs-SS were 356.74 and 342.83 μg/mg, respectively, while those of HT-LMSNs-SS reached 396.79 μg/mg. On the other hand, the EEs% of various vehicles were all above 80%, wherein that of HT-LMSNs-SS was 90.87%, which was slightly higher than that of other nanocarriers. Thus, it could be inferred that besides the inner pores in MSN “cores”, the outer lipid layer of LMSNs could markedly enlarge the loading volumes for drugs, thus being conducive to cargo encapsulation and preventing the premature leakage of drug molecules.

The release rate of DOX from various nano-formulations was investigated under conditions of different pH and GSH concentrations. It should be noted that although the outer lipid layer of LMSNs could help to prevent DOX leakage mainly at static conditions, it did not hinder DOX molecule release under in vitro drug release experimental conditions that simulated in vivo gastrointestinal peristalsis. As shown in Figure 4, MSNs-SS@DOX exhibited more rapid drug release in a pH 5.0 medium than in pH 7.4, and the release rate increased with the increase in GSH concentration. For instance, its maximum release rate was 73.8% in the medium of pH 5.0 and 10 mM of GSH by 96 h. Similarly, a higher release rate in pH 5.0 than in pH 7.4 was observed for other nano-formulations. Furthermore, the cumulative release of DOX from the nano-formulation was relatively slow in the absence of GSH regardless of the pH value. Instead, after the addition of GSH, the DOX was released more rapidly from the nano-formulation, while MSNs@DOX and liposomes@DOX were not included. Therefore, the pH and reduction dual-responsive drug release property of various hybrid LMSNs was confirmed by the above results. The accelerated release of DOX should be attributed to that in the nearly physiological environment. Most pores of MSNs were effectively blocked by -SS-NH_2_ groups, while the lipid layer also acted as an outer barrier to hinder the leakage of drugs. When the pH value declined to 5.0, a weakly acidic condition similar to tumor tissue, the electrostatic interaction between DOX molecules and MSNs was weakened by the protonation effect caused by silanol groups, resulting in the gradual release of DOX. More notably, due to the cleavage of a disulfide bond in the reducing condition with high GSH concentration, the lipid layer would dissociate and detach from the surface of the MSN “core”. In addition, the leaving of -SS-NH_2_ groups greatly favored the free diffusion of entrapped DOX molecules from the pores, leading to rapid drug release.

### 2.3. In Vitro Cytotoxicity

The in vitro cytotoxicity towards MCF-7 and MCF-7/ADR cells of various blank carriers and nano-formulations is shown in Figure 5. After 24 h or 48 h of incubation, the survival rates of MCF-7 cells treated with various carriers all remained higher than 80%, indicating that MSNs and hybrid LMSNs could serve as biocompatible nanocarriers with low cytotoxicity and applicability for further drug delivery application. As for nano-formulations, the time- and concentration-dependent cytotoxicity was observed for both types of cells. Nevertheless, various formulations exhibited different toxic effects on ordinary or resistant tumor cells. For instance, the viability of MCF-7 and MCF-7/ADR cells treated with LMSNs-SS@DOX for 48 h was about 50% and 65%, whereas that of the HT-LMSNs-SS@DOX treatment declined to 24.9% and 25.4%, respectively. The IC_50_ values (the concentrations that inhibited cell growth by 50%) were calculated using SPSS software (version 9.1), and the resistance index (RI) was estimated as IC50 (MCF-7/ADR)/IC_50_ (MCF-7). As listed in Table 3 and Table 4, the IC_50_ values of free DOX against MCF-7/ADR cells at 24 h and 48 h were 159.27 µg/mL and 92.6 µg/mL, respectively, with the RI as high as 5.51 and 5.1, demonstrating the prominent efflux characteristic of highly expressed P-gp in MCF-7/ADR cells. When the cells were treated with nano-formulations, the IC_50_ and RI decreased sharply, which could be explained by the fact that various nanocarriers facilitated the rapid release of DOX and enhanced the cellular uptake efficiency through special endocytic pathways. In particular, HT-LMSNs-SS@DOX showed lower IC_50_ values towards MCF-7/ADR cells than other treatment groups, being only 8.03 and 4.79 µg/mL at 24 h and 48 h, respectively. Therefore, their RI dropped to even below 1.0 (0.99 and 0.95). This result suggested that HT-LMSNs-SS@DOX had the most potent inhibitory activity on MCF-7/ADR cells, which was mainly associated with its capacity for targeted drug delivery and its ability to eliminate the expelling effect of the efflux pump (P-gp) to overcome drug resistance, due to dual modification by HA and TPGS for hybrid LMSN carriers.

### 2.4. Intracellular Uptake and Internalization Mechanism

The cellular uptake of free drugs and nano-formulations was observed using a fluorescence microscope and analyzed through FCM, respectively. As shown in Figure 6A,B, almost no fluorescence signal was detected in cells treated with a control sample (free DOX or FITC), owing to the fact that free drug molecules were transferred into cells only through passive mobility or diffusion. Unlike small molecules, as the nanocarriers, MSNs and hybrid LMSNs were apt to be internalized into cells through an efficient endocytosis pathway. Therefore, various nano-formulations exhibited more efficient cellular uptake, and the fluorescence intensity was increased gradually with the improvement of surface modification for nanocarriers. The merging of different fluorescence implied that many nanocarriers and DOX molecules were already located in the nucleus. Moreover, a similar trend is observed in Figure 7A. The increase of intracellular fluorescence intensity from 4 h to 16 h after treatment verified that various nanocarriers were taken up by cells in a time-dependent manner, while HT-LMSNs-SS-FITC showed a more intense fluorescence signal compared to other carriers, which was approximately 1.58- and 1.64-folds higher than that of free FITC at 4 h and 16 h, respectively.

The internalization mechanism of various nanocarriers was further analyzed by assessing the effect of the energy inhibitor (NaN_3_) or endocytosis inhibitors on their uptake efficiency. Chlorpromazine, MβCD, Dynasore, and EIPA were used as special inhibitors for four typical uptake pathways: the clathrin-related route, the caveolae (or lipid raft)-mediated route, the clathrin- and caveolae/lipid-raft-dependent route, and the macropinocytosis pathway. As shown in Figure 7B, although the energy supplement (primarily in the form of ATP) was essential for the internalization of nanomaterials into cells, temperature and energy were observed to have little impact on the uptake efficiency of various carriers. This result may be attributed to the fact that liposomes and hybrid LMSNs have powerful lipophilicity and entry ability due to the presence of an outer lipid layer, which was conducive to rapid lipid fusing with the cell membrane. The results further demonstrated that the fluorescence intensity in cells was influenced less by the co-incubation with MβCD, Dynasore, or EIPA; in contrast, pretreatment with chlorpromazine resulted in a significant decline in uptake efficiency for each nanocarrier (decreased by approximately 60%). Therefore, it should be inferred that clathrin-related routes were probably the main internalization pathway for various nanocarriers developed in this study.

### 2.5. Cell Apoptosis and Cell Cycle Analysis

To investigate the influence of DOX or various nano-formulations on cell apoptosis and the cell cycle in MCF-7 and MCF-7/ADR cells, an Annexin V-FITC/PI staining assay and a PI single-staining assay were carried out to quantify the apoptosis rate and the variation in cell cycle distribution, respectively.

#### 2.5.1. Apoptosis Analysis through FACS

The cells after treatment were quantitatively distinguished using Annexin V-FITC/PI staining through FACS, which were divided into necrotic cells (Q1, Annexin V-FITC-negative, PI-positive), late apoptotic cells (Q2, both Annexin V-FITC- and PI-positive), early apoptotic cells (Q3, Annexin V-FITC-positive, PI-negative), and normal cells (Q4, both Annexin V-FITC- and PI-negative). After 12 h or 24 h of treatment, the cell apoptotic detection results are shown in Figure 8A,B and Table 5. The total apoptotic ratios of untreated MCF-7 cells were 5% and 4.5% at 12 h and 24 h, while the amounts of apoptotic MCF-7/ADR cells in the control group were 3.69% and 5.09%, respectively. The proportion of total apoptosis of MCF-7 cells induced by free DOX were 21.29% and 30.80% at 12 h and 24 h, respectively, whereas the apoptotic ratios of MCF-7/ADR cells after the same treatment were only 5.57% and 7.45%, respectively, showing the high resistance against chemotherapeutic agents. Moreover, functionalized nano-formulations exhibited a more efficient apoptosis-inducing effect for resistant cells. For example, when MCF-7 and MCF-7/ADR cells were cultured with HT-LMSNs-SS@DOX for 24 h, the proportion of total apoptosis reached 37.1% and 38.25%, respectively. The above results indicated that TPGS conjugated onto the outer lipid layer of LMSNs could be used to inhibit P-gp-mediated drug efflux; as such, HT-LMSNs-SS@DOX complexes might serve as the prominent DDS for controlled drug release with enhanced activity towards resistant tumor cells through the apoptosis pathway. Furthermore, it should be noted that the nano-formulations could induce tumor cells to death through various pathways including necrosis, not only apoptosis.

#### 2.5.2. Cell Cycle Analysis Using FACS

A PI single-staining assay was used to quantitatively analyze the cell cycle ratio of MCF-7 and MCF-7/ADR cells treated with free DOX or nano-formulations. As shown in Figure 9A, the cell cycle percentages of G0/G1 peaks of MCF-7 cells in the control group were 37.7% and 41.0% at 12 h and 24 h, respectively. After treatment with free DOX, the same phase ratios increased to 42.16% and 43.2%. When MCF-7 cells were cultured with HT-LMSNs-SS@DOX for 12 h and 24 h, the G0/G1 phase ratios were rapidly increased to 52.72% and 60.93%, and, simultaneously, the G2/M phase ratios were dropped to 24.53% and 22.51%, respectively. As for the MCF-7/ADR cells, at 12 h and 24 h, the cycle percentages of G0/G1 peaks in the free DOX treatment group were 41.16% and 27.31%, exhibiting no significant difference when compared to the control group (38.06% and 29.00%, respectively, Figure 9B). Nevertheless, after treatment with HT-LMSNs-SS@DOX for 12 h or 24 h, the percentage of G0/G1 peaks markedly increased to 56.35% and 63.90%, while the G2/M peaks declined to 22.40% and 19.52%, respectively. These results were consistent with the cell apoptosis analysis, which implied that HT-LMSNs-SS@DOX complexes could induce apoptosis for both MCF-7 and MCF-7/ADR cells by arresting the G0/G1 phase to a maximum extent [32]. Consequently, HT-LMSNs-SS was a potent delivery vehicle to overcome drug resistance, and it had the potential for inhibiting resistant tumor cell growth and improving the therapeutic effect.

### 2.6. P-gp Expression Detection Using Cellular Immunofluorescence Assay

The overexpression of P-gp is widely regarded as one of the important mechanisms leading to drug resistance in tumor cells. In this study, a fluorescence-labeled secondary antibody (Alexa Fluor 594 goat anti-rabbit IgG) was used to label P-gp protein in MCF-7 cells and MCF-7/ADR cells. As shown in Figure 10A, the red fluorescence intensity of MCF-7/ADR cells was significantly higher than that of MCF-7 cells, which verified the overexpression of P-gp in MCF-7/ADR cells, which was undoubtedly responsible for the development of their drug resistance.

### 2.7. Analysis for Reversal of MDR

The expression of P-gp in MCF-7/ADR cells after 12 h of treatment was detected at the protein level through Western blot analysis. With GAPDH as a reference protein, the expression of P-gp in the control group was the highest (Figure 10B). After treatment with nano-formulations, the bands of P-gp became shallow, and the expression level gradually decreased. In particular, HT-LMSNs-SS@DOX exhibited the most potent inhibitory effect on the expression of P-gp. The results revealed that HT-LMSNs-SS could be effective in overcoming tumor MDR.

The results of the wound healing assay are shown in Figure 10C. In the control group, scratches narrowed notably after 24 h of treatment, which proved that MCF-7/ADR cells were highly active in preventing mechanical damage through rapid migration. Alternatively, the mobility rates of cells treated with DOX, MSNs-SS@DOX, LMSNs-SS@DOX, T-LMSNs-SS@DOX, or HT-LMSNs-SS@DOX were 20%, 21.3%, 18%, 15.8%, and 8%, respectively. The decrease in mobility rates verified the inhibition ability of various nano-formulations on cell migration, while HT-LMSNs-SS@DOX showed the most powerful inhibition effect on cell growth, proved by the variation in scratch width.

### 2.8. Intracellular Tracking in MCF-7/ADR Cells

The endosomal escape of FITC-labeled MSNs-SS and HT-LMSNs-SS in MCF-7/ADR cells was investigated using confocal microscopy. Hoechst and Lyso-Tracker Red were used to label the nucleus and the lysosome, respectively [33]. After co-culturing with MSNs-SS-FITC, the intensity of green fluorescence in cells became intense with the extension of time; however, the overlap degree of green and blue fluorescence was relatively low, even at 6 h (Figure 10D). This result implied that although MSNs-SS nanocarriers could be internalized by cells with time, they could hardly break through the lysosome barrier and enter the nucleus. Conversely, the green fluorescence of HT-LMSNs-SS-FITC almost colocalized with Lyso-Tracker-Red-labeled lysosomes after 1 h of co-incubation (Figure 10E). When incubated for 3 h, it could be observed that HT-LMSNs-SS-FITC gradually translocated from the lysosome to the cytoplasm, and a number of nanocarriers accumulated into the nucleus. Upon extending the incubation time to 6 h, FITC fluorescence overlapped with Hoechst blue fluorescence was higher; meanwhile, the fluorescence intensity of FITC was enhanced. Moreover, the intensity of red fluorescence significantly declined with the increase in incubation time. Therefore, it could be inferred that the outer lipid bilayers of hybrid LMSNs were able to facilitate the carriers’ achievement of lysosomal escape to enter the nucleus more efficiently and exert a curative effect by fusing with the lysosome membrane.

### 2.9. Intracellular ROS Detection of MCF-7/ADR Cells

As one of the important chemical signal molecules, intracellular ROS is normally involved in affecting the physiological and biochemical characteristics of cells and even altering their function. Normally, superfluous ROS is prone to interacting with endogenous DNA or protein, resulting in some downstream signaling pathways being triggered to induce excess cell apoptosis and necrosis. Thus, when programmed cell death occurs, excessive ROS is produced in the mitochondria [34]. After treating with free DOX or various nano-formulations and additional DCFH-DA staining, the ROS levels of MCF-7/ADR cells were quantitatively analyzed by measuring the fluorescence intensity, and the results are shown in Figure 11. It was clearly observed that free DOX only induced tumor cells to produce a small number of ROS; in contrast, the production of ROS in nano-formulation treatment groups notably increased, and the highest fluorescence signal denoting intracellular ROS production was detected in cells treated with HT-LMSNs-SS@DOX at both 12 h and 24 h (Figure 11A,B). In Figure 11C, various nano-formulations exhibit a more remarkable ROS induction effect on resistant tumor cells in a time-dependent manner. Specifically, the ROS level in cells treated with HT-LMSNs-SS@DOX was higher than other treatments, by up to 11.2-fold or 2.4-fold compared to the control and free DOX groups, respectively. These results verified that HT-LMSNs-SS@DOX complexes could induce overproduction of ROS in resistant tumor cells more efficiently to activate the rapid programmed cell death pathways, such as apoptosis.

### 2.10. Assessment of the Antitumor Efficacy In Vivo

Female BALB/c nude mice were randomly divided into five groups: blank control (NS), free DOX, MSNs-SS@DOX, LMSNs-SS@DOX, and HT-LMSNs-SS@DOX. After 15 days of intravenous administration, the tumors were removed and weighed to evaluate the antitumor effect of various treatment groups (Figure 12A).

As shown in Figure 12B, the body weight of mice remained almost constant during treatment for all formulations. The animals receiving treatment with HT-LMSNs-SS@DOX exhibited the largest mean body weight among all tested formulations during 15 days. The tumor volume in the control group continuously increased over time, while the nano-formulation treatments had the more powerful efficacy in retarding tumor growth than free DOX (Figure 12C,F). The better performance of nano-drugs should be attributed to the passive targeting by which more DOX molecules could be delivered and deposited into the tumor site (EPR effect). As expected, HT-LMSNs-SS@DOX was superior over other nano-drug treatments in inhibiting tumor growth (*p* < 0.05). Because the same drug dose was applied for various nano-formulations, the high potency of HT-LMSNs-SS delivery was believed to be a result of dual modification with HA and TPGS and rapid cleavage of the disulfide bond in the tumor microenvironment.

Tumor weight at the end of the in vivo experiment was another important index for efficacy assessment. The average tumor weight in mice treated with HT-LMSNs-SS@DOX was 0.98 ± 0.17 g, which was significantly lower than the treatment of NS (3.08 ± 0.6 g), free DOX (2.76 ± 0.16 g), MSNs-SS@DOX (2.15 ± 0.32 g), and LMSNs-SS@DOX (2.02 ± 0.14 g) (Figure 12E). The superior property of HT-LMSNs-SS vehicles was proved by both in vitro and in vivo results, which could enhance antitumor efficacy towards resistant tumors by boosting responsive drug release and intracellular accumulation after inhibiting the efflux effect of P-gp. As a typical histopathology technique, H&E staining images clearly showed that when compared to the normal organs in the control group, there were no inflammatory lesions or damage to all organs following treatment with free DOX or various nano-formulations (Figure 12G). Moreover, the tumor tissues of mice were collected and subjected to TUNNEL staining to analyze the cell apoptosis. After staining, the nuclei of normal cells and apoptotic tumor cells were blue and brown, respectively. It was apparent that almost no brown nuclei appeared in the NS group; conversely, free DOX or nano-formulations induced cell death to different degrees. The brown areas in tumor sections treated with HT-LMSNs-SS@DOX were the largest among all tested formulations. Likewise, the survival rate of tumor-bearing mice post-formulation treatment was consistent with the above results (Figure 12D). The antitumor potency and the survival period of mice are ranked as follows: HT-LMSNs-SS@DOX > LMSNs-SS@DOX > MSNs-SS@DOX > free DOX. The in vivo experimental results revealed that HT-LMSNs-SS@DOX DDS had good biosafety and that it could induce and accelerate cell necrosis and apoptosis to achieve enhanced therapeutic efficacy against resistant tumors.

## 3. Materials and Methods

### 3.1. Chemicals and Reagents

Doxorubicin hydrochloride (DOX) and fluorescein isothiocyanate (FITC) were purchased from Meilun Biological Technology Co., Ltd. (Dalian, China). Vitamin E polyethylene glycol succinate (TPGS), cetyltrimethylammonium bromide (CTAB, >99%), tetraethyl orthosilicate (TEOS, 98%), diethanolamine (DEA, 99%), and glutathione (GSH) were purchased from Aladdin Chemistry Co., Ltd. (Shanghai, China). Soybean phospholipid (SP) was purchased from Shanghai Tywei Pharmaceutical Co., Ltd. (Shanghai, China). Cholesterol and 1, 2-dipalmitoyl-sn-glycero-3-phospho-(1′-rac-glycerol) (DPPG) were purchased from Avanti Polar Lipids (Alabaster, AL, USA). Hyaluronic acid (HA) was obtained from Macklin Biochemical Technology Co., Ltd. (Shanghai, China). 3-aminopropyltriethoxysilane (APTES), glycolic anhydride (SA), and L-cysteine hydrochloride anhydrous substance (Cys·HCl) were purchased from Aladdin Chemistry Co., Ltd. (Shanghai, China).

1-(3-Dimethylaminopropyl)-3-ethyl-carbo-diimide hydrochloride (EDC·HCl), N-hydroxy-succinimide (NHS), 3-(4,5-dimethylthiazol-2-yl)-2,5-diphenyltetrazolium bromide (MTT), the Roswell Park Memorial Institute 1640 (RPMI-1640), 0.25% (*w*/*v*) trypsin-0.03% (*w*/*v*) EDTA solution, and phosphate buffer solution (PBS) were all purchased from Solarbio Biochemical Technology Co., Ltd. (Beijing, China). Fetal bovine serum (FBS), penicillin/streptomycin solution, and DAPI staining solution were obtained from Tianhang Biotechnology Co., Ltd. (Hangzhou, China). The Annexin V-FITC Apoptosis Detection Kit, Cell Cycle Analysis Kit, ROS Assay Kit, and Hoechst 33342 nuclear staining solution were purchased from Beyotime Biological Technology Co., Ltd. (Shanghai, China). Chlorpromazine hydrochloride was purchased from Macklin Biochemical Co., Ltd. (Shanghai, China). Methul-β-cyclodextrin (MβCD) and Dynasore were purchased from Meilun Biological Technology Co., Ltd. (Dalian, China). 5-(N-ethyl-N-isopropyl) amiloride (EIPA) was obtained from Toronto Research Chemicals Inc. (North York, ON, Canada). Sodium azide (NaN_3_) was purchased from Solarbio Biochemical Technology Co., Ltd. (Beijing, China).

Ultrapure water was produced using a UP Water Purification System (Ningbo, China). Ammonium nitrate, chloroform, and sodium dihydrogen phosphate were purchased from Haohua Chemical Reagent Co., Ltd. (Luoyang, China). Chromatographically pure methanol and ethanol were purchased from Shield Specialty Chemical Co., Ltd. (Tianjin, China). P-gp primary antibody, GAPDH primary antibody, and goat anti-rabbit secondary antibody were purchased from Proteintech Group, Inc. (Wuhan, China) and LI-COR Biosciences (Lincoln, NE, USA), respectively. All other chemicals were analytical-grade and used without any further purification.

### 3.2. Cell Culture

The human breast cancer cell line MCF-7 and its drug-resistant strain MCF-7/ADR used in this study were obtained from American Type Culture Collection (ATCC, Manassas, VA, USA) and cultured in RPMI 1640 medium containing 10% (*v*/*v*) FBS and 1% penicillin–streptomycin at 37 °C in a humidified atmosphere with 5% CO_2_.

### 3.3. Experimental Animals

All animal experiments followed the guidelines of the National Institutes of Health and the rules and guidelines approved by the Ethics Committee of the Animal Center of Zhengzhou University. Female BALB/c nude mice of 4–6 weeks of age (body weight: 15–20 g) were purchased from Beijing WeitongLihua Experimental Animal Technology Co, Ltd. MCF-7/ADR cells (1 × 10^7^) and 0.2 mL of normal saline (NS) were subcutaneously injected (s.c.) into the right forelimb of the mice to establish the MCF-7/ADR tumor-bearing nude mouse model.

### 3.4. Synthesis of MSNs and Preparation of Hybrid and Functionalized LMSNs

The MSNs were synthesized through the template agent method according to our new approach [24,35]. Then, 100 mg of MSNs was dispersed in 10 mL of anhydrous ethanol and churned for 1 h. The 2 mL of APTES was appended dropwise into the above solution under intensive stirring. The refluxing reaction was maintained at 60 °C for 20 h [36]. The precipitate was collected through centrifugation at 11,000 rpm and further dried under vacuum for 24 h to obtain MSNs-NH_2_.

The 10 g of SA was dissolved in acetone, and 100 mg of MSNs-NH2 was dispersed in 10 mL of acetone, respectively [37]. The MSNs-NH_2_ solution was stirred for 1 h; afterwards, 5 mL of SA solution was added dropwise. After another 24 h of stirring at room temperature, the resulting solution was centrifugated, and the precipitate was washed with anhydrous ethanol three times. MSNs-COOH were collected after vacuum drying for 24 h [38].

Next, 100 mg of MSNs-COOH was redispersed in 20 mL of PBS buffer (pH 5.0). Then, EDC·HCl and NHS were added into the above suspension and stirred for 2 h. To introduce the thiol group, 1 g of Cys·HCl was allowed to react with MSNs-COOH at 35 °C under a nitrogen atmosphere [39]. After the disulfide bond exchange reaction for 24 h, the final nanoparticles were separated through centrifugation and vacuum drying overnight, named MSNs-SS-NH_2_ (MSNs-SS).

Liposomes were prepared through a film dispersion method. SP, cholesterol, and DPPG were dissolved in 10 mL of chloroform under ultrasonic dispersion. Subsequently, the lipid mixture was evaporated under a vacuum at 65 °C to form a dried film, and N_2_ gas was used to remove any residual solvent. The lipid film was then resuspended with ultrapure water and suspended through sonication for 10 min. The lipid suspension was further disrupted with an ultrasonic probe, and the resulting liposomes were sterilized through extrusion using a sterile filter.

MSNs-SS were dispersed in ultrapure water under sonication. The lipid film was produced using the film dispersion method, as described above, and further hydrated with MSNs-SS suspension through continuous sonication, to obtain the hybrid LMSN samples (LMSNs-SS). The 50 mg of TPGS and various lipid materials were dissolved in 10 mL of chloroform, and the MSNs-SS suspension was similarly used to hydrate the above lipid mixture to obtain the TPGS-modified LMSN samples (T-LMSNs-SS).

Lastly, 50 mg of HA was hydrated in ultrapure water overnight and then activated with EDC·HCl and NHS through 2 h of stirring. The T-LMSNs-SS suspension was added, and the mixture was immediately churned for 24 h. Repeated centrifugation was performed to remove the excess HA. The precipitate was resuspended with ultrapure water, and the final multi-functionalized LMSN samples (HT-LMSNs-SS) were obtained through freeze-drying.

### 3.5. Labeling Using FITC and Loading of DOX for Nanocarriers

The 100 mg of MSNs or MSNs-SS was dispersed in ultrapure water. After 1 h of sonication, FITC solution (5 mg/mL, dissolved in DMSO) was appended dropwise. The mixture was then stirred under the lightproof condition for 24 h. The final FITC-labeled MSN samples (MSNs-FITC or MSNs-SS-FITC) were produced through centrifugation and redispersed in a small amount of ultrapure water.

The lipid film was prepared as described above. Then, the FITC solution (dissolved in ultrapure water) was added to hydrate lipid film to obtain the final FITC-labeled liposomes (liposomes–FITC). Afterward, MSNs-SS-FITC suspension was used to hydrate the lipid film to produce LMSNs-SS-FITC, T-LMSNs-SS-FITC, and HT-LMSNs-SS-FITC, followed by the same procedures as described above.

The 100 mg of MSNs was dispersed in ultrapure water; after 1 h of sonication, the DOX aqueous solution was appended slowly. The mixture was then stirred at room temperature for 24 h. After centrifugation at 11,000 rpm for 15 min, the final DOX-loaded MSN samples (MSNs@DOX) were produced by vacuum drying overnight. MSNs-SS@DOX were prepared by linking the disulfide bonds onto MSNs@DOX samples. Briefly, EDC·HCl and NHS were added into the MSNs@DOX dispersion and stirred. Then, MSNs@DOX was reacted with Cys·HCl under a nitrogen atmosphere for 24 h. After the disulfide bond exchange reaction, the MSNs-SS@DOX were separated and collected through centrifugation and vacuum drying overnight.

As described above, the DOX aqueous solution was used to hydrate the lipid film to obtain the DOX-loaded liposomes (Liposomes@DOX). Similarly, various DOX-loaded LMSN samples (LMSNs-SS@DOX, T-LMSNs-SS@DOX, and HT-LMSNs-SS@DOX) were prepared using MSNs-SS@DOX. Briefly, MSNs-SS@DOX was redispersed, and the suspension was used to hydrate the lipid film to produce LMSNs-SS@DOX. TPGS and HA were activated and further conjugated onto LMSNs-SS@DOX by following the same procedures described above. The precipitate was resuspended, and the final HT-LMSNs-SS@DOX samples were obtained through freeze-drying. All of the samples were prepared the day before the experiment and stored at 4 °C. 

### 3.6. Characterization of Various Nanocarriers

The sample morphology of various nanocarriers was observed through transmission electron microscopy (TEM, HICHI) and scanning electron microscopy (SEM, SU8020). For TEM observation, liposome samples were negatively stained with 1% phosphotungstic acid. The diameter size and zeta potential of nanocarriers were measured through dynamic light scattering (Nano ZS90, Malvern, UK). Fourier transform infrared (FT-IR) spectra of various nanocarriers and DOX-loaded samples were recorded using an FT-IR spectrometer (Nicolet IS10). An X-ray diffraction (XRD) diffractometer was used to detect the material composition of various nanocarriers (BRUCKER D8) over the 2θ range of 3°–60° at a scanning rate of 5°/min. The elemental analysis of C, O, S, and Si in various nanocarriers was determined through X-ray photoelectron spectroscopy (XPS, Thermo Fisher Scientific, Waltham, MA, USA). After placing them under the objective of the microscope and focusing, the Raman spectra of various dried samples were recorded (laser excitation wavelength set at 514 nm, 500–4000 cm^−1^). The specific surface area, porosity, and pore size of various nanocarriers were measured using an adsorption analyzer (ASAP 2460), and nitrogen adsorption–desorption isotherms were obtained at −180 °C under continuous adsorption conditions.

### 3.7. In Vitro Drug Release

The cumulative release rates of DOX from different nano-formulations were examined through a dialysis method. A PBS-anhydrous ethanol mixed solution (1:2, *v*/*v*) was prepared as the release medium, and the release profile of various formulations was assessed at pH 7.4 or 5.0 and different GSH concentrations, respectively. In brief, free DOX aqueous solution or DOX-loaded nano-formulations were dispersed in 5 mL of medium (5 mg/mL) and then transferred to a dialysis bag (MWCO: 8000–14,000 Da). Next, various samples were suspended in 50 mL of PBS under continuous shaking at a speed of 100 rpm at 37 °C. At preset time intervals, 2 mL of supernatant outside of the dialysis bag was taken out and replaced with the same volume of fresh PBS. The concentration of released DOX was determined through HPLC assay. The cumulative release rate of DOX was calculated, measurements were repeated three times, and the results were averaged.

HPLC Assay. The chromatography separation was performed with a DiamonsilTMC_18_ Column (200 mm × 4.6 mm, 5 µm); the flow rate was 1.0 mL/min, the column temperature was 25 °C, and the mobile phase was methanol–sodium dihydrogen phosphate (1:1). DOX was quantitatively analyzed at a maximum absorption wavelength of 480 nm [40].

### 3.8. Cell Viability

The cytotoxicity of various blank nanocarriers and their DOX-loaded formulations against MCF-7 or MCF-7/ADR cells was assessed through MTT assay. The MCF-7 or MCF-7/ADR cells were cultivated in 96-well plates at a density of 7 × 10^4^ cells/mL and incubated overnight. Then, the culture medium was removed and replaced with fresh medium containing different blank nanocarriers (10–200 μg/mL), free DOX, or nano-formulations (2.5–20 μg/mL of DOX concentration), respectively. After 24 h or 48 h, 20 μL of MTT solution (5 mg/mL) was added and further incubated for 4 h. Subsequently, the medium was removed, and 100 μL of DMSO was added to disperse the formazan precipitate. The multi-function plate reader (Varioskan Flash, Thermo Fisher Scientific) was used to measure the optical density of the solution at 490 nm [41,42].

### 3.9. Cellular Uptake and Internalization Mechanism Analysis

The cellular uptake of various nano-formulations was determined through flow cytometry (FCM). MCF-7/ADR cells were cultured overnight at a density of 3 × 10^5^ cells per well in 6-well plates. After exposure to each formulation (5 μg/mL of DOX concentration) for different times (4 and 16 h), the cells were washed twice with PBS and then collected through centrifugation at 1000 rpm for 5 min. Lastly, the cells were resuspended in PBS, and the uptake of various nanocarriers was quantified using FCM [43]. In addition, the cellular uptake was observed using fluorescence detection technology. After trypsinizing and staining with Hoechst 33342 for 30 min, the cells were rinsed with PBS twice and immediately fixed with paraformaldehyde (4%, *w*/*w*). A fluorescence microscope was used to detect the fluorescence signal inside the cells [44].

The internalization mechanism of various nanocarriers was comparatively analyzed by examining their uptake efficiency variation after treatment with the energy inhibitor or endocytosis inhibitors. MCF-7/ADR cells were incubated at 4 °C or 37 °C or treated with 200 μL of energy inhibitor (NaN_3_). After 2 h of incubation, the medium was removed and replaced with a fresh serum-free medium containing free FITC or nanocarriers (labeled by FITC, 10 μg/mL of FITC concentration) for an additional 4 h of incubation at 37 °C. Following the same procedures, four endocytosis inhibitors (chloropromazine, MβCD, EIPA, or Dynasore; each inhibitor was adjusted to the optimal concentration as required before the experiment) were also used to pretreat the cells. The cells were then trypsinized and rinsed with cold PBS. Finally, the total fluorescence intensity within cells was measured using FCM after resuspending the cell suspension [45]. 

### 3.10. Cell Apoptosis and Cell Cycle Assay

The MCF-7 cells and MCF-7/ADR cells were seeded in 6-well plates overnight at a density of 3 × 10^5^ cells per well at 37 °C. Then, the cells were treated with free DOX or various nano-formulations at 5 μg/mL of DOX concentration. After 12 h or 24 h of incubation, the cells were washed with PBS 2–3 times, trypsinized, and collected through centrifugation. Afterward, 500 μL of buffer solution containing Annexin V-FITC and PI was added for further incubation in darkness. The cell apoptosis ratio was determined through Fluorescence-Activated Cell Sorting (FACS). The cells without any treatment were used as the control group [46].

The experimental procedure for cellular cycle measurement was similar to the above process. The collected cells were resuspended in 70% cold ethyl alcohol overnight to avoid structural damage to the cytomembrane [47]. The cell suspensions were subsequently centrifugated to remove ethyl alcohol, and 500 μL of PI buffer solution was added for incubation at 37 °C in darkness. The cells without any treatment were used as the control group to analyze the cell cycle variation.

### 3.11. Cellular Immunofluorescence

The MCF-7 and MCF-7/ADR cells were seeded in 24-well plates and fixed with 500 µL of 4% paraformaldehyde for 30 min at 4 °C. After permeating with 500 µL of 0.3% Triton X-100 and washing with PBS for three times, the cells were treated with blocking buffer containing 3% of BSA under shaking. The P-gp primary antibody and goat anti-rabbit secondary antibody (labeled by Alexa Fluor 594) were added for 18 h and 1 h of incubation at 4 °C, respectively. Lastly, the cells were stained with Hoechst 33342 at room temperature for 10 min, washed with PBS, observed using a fluorescence microscope, and photographed [35].

### 3.12. Western Blot Analysis and Wound Healing Assay

The MCF-7/ADR cells were cultured in 6-well plates and then treated with free DOX or various nano-formulations at 5 μg/mL of DOX concentration for 12 h. Total cellular proteins were obtained by incubating the cells in lysis buffer on ice and a subsequent metal bath at 37 °C for 30 min. The cell lysates were separated through electrophoresis in 7.5% of polyacrylamide gels and further transferred to NC membranes. The membranes were closed with a TBS solution containing 5% skim milk powder for 1 h and subsequently washed with TBST four times [48]. The expression levels of P-gp were examined through Western blotting with an infrared laser imaging system, using glyceraldehyde-3-phosphate dehydrogenase (GAPDH) as the internal standard.

The MCF-7/ADR cells were cultured in 24-well plates, and then 200 μL of the suction tip was used to scratch through the bottoms of the plates to cause a vertical wound. Then, the culture medium was removed, and the cells were washed with PBS. Next, 500 μL of fresh medium was subsequently added per well, and the scratch area was placed in the center of the field of view and observed using an inverted optical microscope. The cells were incubated with DOX or various nano-formulations at 5 μg/mL of DOX concentration for another 24 h. The scratch width variation at different time points was recorded to calculate the cell migration rate.

### 3.13. Intracellular ROS Detection

The cell culture and sample treatments were similar to the procedure described in 2.10. After incubation for 2 h, 6 h, 12 h, and 24 h, 1 mL of RPMI-1640 medium (containing 10 μM of DCFH-DA) without FBS was added into each well. The cells were incubated for another 30 min and then washed with fresh medium 2–3 times, trypsinized, and collected through centrifugation. Lastly, the cell suspension resuspended in 500 μL of cold PBS was used to detect intracellular ROS levels using the FACS [49]. Fluorescence images were obtained at a specific time using a fluorescent microscope.

### 3.14. In Vivo Antitumor Effect Evaluation

To evaluate the in vivo antitumor effect and biosafety of various nano-formulations, MCF-7/ADR tumor-bearing nude mice received intravenous administration of free DOX or nano-drugs. All of the nano-formulations were sterilized using steam under high pressure. When the tumor volume reached 60–90 mm^3^, the nude mice were randomly divided into five groups. The mice in each group were administered via the tail vein an injection with NS, DOX, MSNs-SS@DOX, LMSNs-SS@DOX, or HT-LMSNs-SS@DOX (all at 3 mg/kg/d of DOX concentration) [50]. After continuous dosing for 15 days, the antitumor effect of each formulation was evaluated by examining tumor volume and body weight every 3 days. The tumor volume was calculated according to the following formula: tumor volume = (length × width^2^)/2 [51]. Also, the survival rate of mice post-formulation treatment over 40 days was recorded to plot the Kaplan–Meier survival curve.

Lastly, the mice were sacrificed to remove the tumor and major organs, and the resected tumor was weighed and photographed. The major organs and tumor tissues after perfusion were fixed overnight in 4% paraformaldehyde. The tissues were soaked in a mixture of xylene and molten paraffin (volume ratio 1:1) and soaked overnight. The organized wax blocks were sliced with a slicer and affixed to the slide. After dewaxing, the wax blocks were subjected to hematoxylin and eosin (H&E) staining or TUNEL staining.

### 3.15. Statistical Analysis

All data are prepared as mean ± SD, and the mean values were considered significantly different when * *p* < 0.05, ** *p* < 0.01, or *** *p* < 0.001. ANOVA was used for statistical analysis of all treatment groups using SPSS software [52].

## 4. Conclusions

In summary, the hybrid MSN-supported lipid bilayer vehicles were designed as targeted delivery nanosystems against drug-resistant tumors, wherein the -SS-NH_2_ was linked onto inner MSNs as “gatekeepers” for the drug molecules inside the pores, and TPGS and HA were conjugated onto the outer lipid bilayer. The obtained functionalized LMSN vehicles were demonstrated to possess better modification and higher drug-loading efficiency through various characterization methods. The functionalized HT-LMSNs-SS@DOX showed dual pH/reduction responsive drug release characteristics. In vitro experimental results revealed the more efficient antitumor activity of HT-LMSNs-SS@DOX compared to other DDSs, which could induce rapid apoptosis of both MCF-7 and MCF-7/ADR cells, arrest cell cycle at the G0/G1 phase, and cause tumor cells to produce excessive ROS. Consistently, HT-LMSNs-SS@DOX was potent in reversing drug resistance by inhibiting the expression of P-gp and facilitating efficient cellular uptake and subsequent lysosomal escape, which was the main reason for their enhanced efficacy against resistant tumors. In addition, the in vivo antitumor efficacy of various LMSN DDSs was evaluated in BALB/c model nude mice. Therefore, the multi-functionalized LMSNs developed in current work could serve as a novel nano-delivery platform and hold value for a diverse range of applications in the treatment of resistant solid tumors.

## Data Availability

Data are contained within the article.

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
