# Peer review of "Enhanced Efficacy against Drug-Resistant Tumors Enabled by Redox-Responsive Mesoporous-Silica-Nanoparticle-Supported Lipid Bilayers as Targeted Delivery Vehicles"

_ijms, 2024, doi:10.3390/ijms25105553_

Round 1

Reviewer 1 Report

Comments and Suggestions for Authors

Very nicely done research work on the application of MSNs.

The research address the purpose for the development of nano-carriers to target the drug resistant tumors.

The multi-functionalized LMSNs created in this work have potential uses in the treatment of refractory solid tumours and could be a unique nano-delivery platform.

Authors have nicely written the conclusion and the same is in align with the presented arguments.

Appropriate references were incorporated in the manuscript.

However, a small concern is in relation to the sterilization of final multi-functionalized MSNs formulation as it was injected subcutaneously.

Hence, it is important to include a part on sterilization of final multi-functionalized MSNs formulation in materials and method section.

Improve the quality of the figures 2, 3, 4, 5, 6, 7 and 9.

Author Response

Manuscript ID: IJMS-2980069

Title: Enhanced efficacy against drug-resistant tumors enabled by redox-responsive mesoporous silica nanoparticles-supported lipid bilayers as targeted delivery vehicles

Dear reviewer,

We are very thankful for the invaluable comments we have received from you. We have therefore revised the manuscript in keeping with the comments below. Taking all your comments, we have revised the manuscript accordingly, labeled them in green color as highlight. Our responses immediately follow each comment.

Thanks again for that you have generously shared your time and professional expertise to help us improve this paper.

Sincerely,

Shuoye Yang,

College of Bioengineering, Henan University of Technology, Zhengzhou 450001, P. R. China,

  1. However, a small concern is in relation to the sterilization of final multi-functionalized MSNs formulation as it was injected subcutaneously. Hence, it is important to include a part on sterilization of final multi-functionalized MSNs formulation in materials and method section.

Response: Thanks for the reviewer’s reminder. A part on the sterilization of nano-formulations was added in the revised manuscript.

  1. Improve the quality of the figures 2, 3, 4, 5, 6, 7 and 9.

Response: Thanks for the reviewer’s suggestion. These figures were redrawn.

Reviewer 2 Report

Comments and Suggestions for Authors

The authors present a multi-component NP system to target multi-drug resistant tumors. MSNs are used as template, onto which disulfides (SS), DOX, liposome layer (L), hyaluronic acid (HA, or H), and tocopherol peg succinate (TPGS, or T) are added, resulting in a HT-LMSNs-SS@DOX NP. The separate layers and final product are characterized and tested toward MCF-7/ADR cells, both in vitro and in vivo models, showing their improved targeting. If the following issues are addressed, the manuscript should be considered for publication.

·       The authors should refer to their recent work, where TPGS and lipid coating methods are also used (S. Yang – 2023: https://doi.org/10.1016/j.jddst.2023.104243 ). Referencing should be made in the method section and in the introduction, giving context to the current manuscript.

·       The introduction needs to go more into LMSN systems and why this system is chosen for this application. Please add references to other papers using this kind of system. The same comment for the lipid layer, is there a reason to use this specific mixture of lipids/compounds?

·       It is unclear when the DOX is introduced in the fabrication of the NPs. Scheme 1 shows DOX loading prior to SS addition, Line 133-132 suggests DOX is added after HA/TPGS functionalization, Line 171 suggests prior to lipid coating (but does not specify further), Line 608-612 says DOX after MSN-SS, Line 613-617 reads as if DOX-containing lipid films are used to create the LMSN@DOX.

·       Please be consistent with name giving, it is very hard to determine whether the author is discussing DOX-loaded or empty NPs. Also, refrain from using: nanocarrier, DDS, NDDS, NP, carrier-formulation, nano-formulation, and sample in the text. Choose one way and change the rest. To improve clarity, it is also better to specify the NPs in the manuscript/figure caption than to just say ‘various DDSs were tested’.

·       Fig 1C does not show a bilayered liposome as one would expect? How were the liposomes imaged? How did you fix/stain them beforehand? For liposomes, normally you would do cryo-TEM due to their instability.

·       For Fig 1D-F, it seems that after lipid coating there are not single NPs anymore but rather clusters are formed? This could explain the size increase shown in Table 1, where a 60 nm increase is shown rather than a 3-7 nm increase. Also, after SS and L addition, the PDI becomes > 0.2, indicating aggregates/clusters/non-uniformity? Please discuss these observations (DLS vs TEM) in the Result section.

·       In Table 2, you are expecting more layers are preventing DOX leakage, but 4 C-F shows the more layers the higher the release rate is (under all conditions). How do you explain this difference?

·       For Fig 3A and 3F, in ‘sample f’ the liposome signals seem to have disappeared? Please add an explanation/discussion in your manuscript.

·       The method section says the final multi-functionalized LMSN samples (HT-LMSNs-SS) were obtained by freeze-drying. Did you check the stability/completeness of the HT-L film after rehydration, especially upon exposure to in vitro and in vivo conditions?

·       For section 2.4 and onwards, what is the concentration of NPs used? How did you determine the concentration to add, is it based on the DOX concentration, or is it based on the how NP weight? (only Section 3.10 and 3.12 mentions partly how it is done) Section 3.8 and onwards is missing the volumes and concentrations of samples used.

·       Line 736-737. How come 3 mg/kg DOX and 10 mg/kg NPs are used? How is this a fair comparison? This needs to be discussed.

Smaller comments:

·       Add references to Line 88-90, 96-98, 100-102, 103-106, and 122-124.

·       Line 129-132, please check if the reference is correct. Ref 45 is about ABC transporters?

·       Fig 1 and Fig 2 are missing referencing to their subfigures in the main text. Also referencing of Fig 10A is missing.

·       Fig 1 A-F should have same scale bars for easy comparison.

·       Fig 3. Annotate important peaks and add chemical bonds in the figure to make it easier to read.

·       Line 197-199, Fig 3C. Please annotate the peaks in the figure, it is not clear now.

·       Fig 6A-B are much too small to see any of the observations mentioned, consider putting in separate figure.

·       Fig 6C names on x-axis are not comparable to names in A-B? Also FI of what; FITC, DAPI, DOX?

·       Fig 7 should be redone so the text and percentages within can be read. Also explanation for the colors used in 7C-D should be added.

·       General for all cell imaging from Fig6-9, please explain all dyes/target compounds/colors in the figure caption.

Comments on the Quality of English Language

The abstract is inconsistent with its abbreviations, please check carefully. 

Please try improving the English in the introduction (the rest of the manuscript is okay). There is a lot of long sentences with a lot of commas, try dividing those in multiple sentences. It may help with the flow of the introduction.

Author Response

Manuscript ID: IJMS-2980069

Title: Enhanced efficacy against drug-resistant tumors enabled by redox-responsive mesoporous silica nanoparticles-supported lipid bilayers as targeted delivery vehicles

Dear reviewer,

We are very thankful for the invaluable comments we have received from you. We have therefore revised the manuscript in keeping with the comments below. Taking all your comments, we have revised the manuscript accordingly, labeled them in green color as highlight. Our responses immediately follow each comment.

Thanks again for that you have generously shared your time and professional expertise to help us improve this paper.

Sincerely,

Shuoye Yang,

College of Bioengineering, Henan University of Technology, Zhengzhou 450001, P. R. China,

  1. The authors should refer to their recent work, where TPGS and lipid coating methods are also used (S. Yang-2023: https://doi.org/10.1016/j.jddst.2023.104243 ). Referencing should be made in the method section and in the introduction, giving context to the current manuscript.

Response: Thanks for the reviewer’s suggestion. Our this work was added as a new reference (reference 24).

  1. The introduction needs to go more into LMSN systems and why this system is chosen for this application. Please add references to other papers using this kind of system. The same comment for the lipid layer, is there a reason to use this specific mixture of lipids/compounds?

Response: Thanks for the reviewer’s suggestion. The merits of LMSN delivery systems were described in detail, and this introduction was added in the revised manuscript. Another reference about this aspect was also added (reference 25).

  1. It is unclear when the DOX is introduced in the fabrication of the NPs. Scheme 1 shows DOX loading prior to SS addition, Line 133-132 suggests DOX is added after HA/TPGS functionalization, Line 171 suggests prior to lipid coating (but does not specify further), Line 608-612 says DOX after MSN-SS, Line 613-617 reads as if DOX-containing lipid films are used to create the LMSN@DOX.

Response: Thanks for the reviewer’s question. The DOX loading was performed prior to SS addition, the regarding description was revised in section 3.5. As to the description in Line 135-138 (former Line 133-132), we only introduce how the drug DOX was loaded by carriers, not meaning that DOX is added after HA/TPGS functionalization.

  1. Please be consistent with name giving, it is very hard to determine whether the author is discussing DOX-loaded or empty NPs. Also, refrain from using: nanocarrier, DDS, NDDS, NP, carrier-formulation, nano-formulation, and sample in the text. Choose one way and change the rest. To improve clarity, it is also better to specify the NPs in the manuscript/figure caption than to just say ‘various DDSs were tested’.

Response: Thanks for the reviewer’s suggestion. The giving names for various samples were revised uniformly, and the “nanocarriers” and “nano-formulations” were referred to the empty and DOX-loaded NPs, respectively.

  1. Fig 1C does not show a bilayered liposome as one would expect? How were the liposomes imaged? How did you fix/stain them beforehand? For liposomes, normally you would do cryo-TEM due to their instability.

Response: Thanks for the reviewer’s question. Liposomes were negatively stained with 1% phosphotungstic acid beforehand, and this introduction was added in the revised manuscript. Indeed, the thin bilayer could be observed for liposomes.

  1. For Fig 1D-F, it seems that after lipid coating there are not single NPs anymore but rather clusters are formed? This could explain the size increase shown in Table 1, where a 60 nm increase is shown rather than a 3-7 nm increase. Also, after SS and L addition, the PDI becomes > 0.2, indicating aggregates/clusters/non-uniformity? Please discuss these observations (DLS vs TEM) in the Result section.

Response: Thanks for the reviewer’s question. The regarding description for size increase and PDI variation for LMSN samples was added in the revised manuscript.

  1. In Table 2, you are expecting more layers are preventing DOX leakage, but 4 C-F shows the more layers the higher the release rate is (under all conditions). How do you explain this difference?

Response: Thanks for the reviewer’s question. For the LMSN samples, the layers of HT-LMSNs-SS were not more than LMSNs-SS and T-LMSNs-SS, only HA and TPGS were conjugated onto the surface of LMSNs. Indeed, the outer lipid layer could help to prevent DOX leakage mainly at static conditions, but it would not hinder DOX molecules release under in vitro drug release experiment conditions that simulate in vivo gastrointestinal peristalsis. The regarding description was added in the revised manuscript.

  1. For Fig 3A and 3F, in ‘sample f’ the liposome signals seem to have disappeared? Please add an explanation/discussion in your manuscript.

Response: Thanks for the reviewer’s question. The regarding discussion was added in the revised manuscript.

  1. For section 2.4 and onwards, what is the concentration of NPs used? How did you determine the concentration to add, is it based on the DOX concentration, or is it based on the how NP weight? (only Section 3.10 and 3.12 mentions partly how it is done) Section 3.8 and onwards is missing the volumes and concentrations of samples used.

Response: Thanks for the reviewer’s reminder and suggestion. In cellular uptake experiment, we determined the concentration to add was based on the DOX concentration, and in internalization mechanism analysis experiment, we determined the concentration to add was based on the FITC concentration. The concentrations of samples used in Section 3.8, 3.9 and 3.12 were added in the revised manuscript.

  1. Line 736-737. How come 3 mg/kg DOX and 10 mg/kg NPs are used? How is this a fair comparison? This needs to be discussed.

Response: Thanks for the reviewer’s question. The description about concentration used in in vivo experiment was indeed incorrect, the concentration used for various formulations including DOX solution were all at 3 mg/kg/d of DOX concentration. The regarding discussion was corrected in the revised manuscript.

  1. Line 129-132, please check if the reference is correct. Ref 45 is about ABC transporters?

Response: Thanks for the reviewer’s reminder. Ref 45 used here was indeed improper, a new reference (reference 47) was added to replace in the revised manuscript.

  1. Fig 1 and Fig 2 are missing referencing to their subfigures in the main text. Also referencing of Fig 10A is missing.

Response: Thanks for the reviewer’s suggestion. The subfigures of these figures were referenced in the revised manuscript.

  1. Fig 1 A-F should have same scale bars for easy comparison.

Response: Thanks for the reviewer’s suggestion. I’m sorry to claim that in the TEM observation experiments, many images in 100 nm and 200 nm scales were not clear, thus the clear images in different scales were selected.

  1. Fig 3. Annotate important peaks and add chemical bonds in the figure to make it easier to read.

Response: Thanks for the reviewer’s suggestion. The important peaks were annotated and the chemical bonds were added in Figure 3.

  1. Line 197-199, Fig 3C. Please annotate the peaks in the figure, it is not clear now.

Response: Thanks for the reviewer’s suggestion. The important peaks were annotated in Figure 3C.

  1. Fig 6A-B are much too small to see any of the observations mentioned, consider putting in separate figure.

Response: Thanks for the reviewer’s suggestion. Figure 6A-B were redrawn as a new separate figure.

  1. Fig 6C names on x-axis are not comparable to names in A-B? Also FI of what; FITC, DAPI, DOX?

Response: Thanks for the reviewer’s question. Indeed, the samples used to treat cells in Figure 6A-B were the DOX-loaded nano-formulations, whereas the ones used to treat cells in Figure 6C were the FITC-labelled nanocarriers. These were specified in the figure caption in the revised manuscript.

  1. Fig 7 should be redone so the text and percentages within can be read. Also explanation for the colors used in 7C-D should be added.

Response: Thanks for the reviewer’s suggestion. Figure 7 was redrawn and split into two separate figures. The explanation for the colors used in Figure 7C-D was added in the figure caption.

  1. General for all cell imaging from Fig 6-9, please explain all dyes/target compounds/colors in the figure caption.

Response: Thanks for the reviewer’s suggestion. The dyes/target compounds/colors were added in the figure caption.

Comments on the Quality of English Language:

  1. The abstract is inconsistent with its abbreviations, please check carefully. 

Response: The abbreviation in abstract section was checked and corrected.

  1. Please try improving the English in the introduction (the rest of the manuscript is okay). There is a lot of long sentences with a lot of commas, try dividing those in multiple sentences. It may help with the flow of the introduction.

Response: Thanks for the reviewer’s suggestion. The long sentences in introduction section were checked and rewritten into shorter multiple sentences.

Reviewer 3 Report

Comments and Suggestions for Authors

Dear Authors,

I have reviewed your manuscript and would like to provide the following comments and suggestions:

1. The chemical reaction and conjugation process for the synthesis of MSNs-SS@DOX and HT-LMSNs-SS@DOX could be elaborated further for better clarity.

2. What is the reasons behind the higher drug loading and encapsulation efficiency observed for Liposomes@DOX compared to MSNs-SS@DOX nanoparticles.

3. Considering redesigning the abstract using biorender or similar software for better visual representation.

4. Please provide the concentrations used for the free drug and drug-loaded MSNs in the cell viability, apoptosis, cell uptake and in cell cycle arrest studies mentioned in the results section.

5. While the IC50 values for LMSNs-SS@DOX are significantly different from MSNs@DOX, the same is not observed for LMSNs-SS@DOX and T-LMSNs-SS@DOX or HT-LMSNs-SS@DOX Vs. Please comment for this observation.

6. Please include statistical analysis for the IC50 data.

7. Since doxorubicin exhibits autofluorescence, it is surprising that the drug's fluorescence is not visible in the fluorescence microscopy imaging. Please provide your thoughts on this observation.

8. At the 16-hour time point of HTMSNs for cell uptake, most cells appear viable, yet you mentioned that the HTMSNs is significantly effective in  killing more number of cancer cells at 24 hours treatment compared to free drug. Please clarify.

9. The apoptosis data shows that the ratio of apoptotic cells for MSNs and L-MSNs is not significantly different. However, the IC50 values indicate a significant difference in cytotoxicity between these two nanoparticle formulations when treated on cells. This seeming contradiction between the apoptosis assay and IC50 data for MSNs and L-MSNs needs to be addressed.

Author Response

Manuscript ID: IJMS-2980069

Title: Enhanced efficacy against drug-resistant tumors enabled by redox-responsive mesoporous silica nanoparticles-supported lipid bilayers as targeted delivery vehicles

Dear reviewer,

We are very thankful for the invaluable comments we have received from you. We have therefore revised the manuscript in keeping with the comments below. Taking all your comments, we have revised the manuscript accordingly, labeled them in green color as highlight. Our responses immediately follow each comment.

Thanks again for that you have generously shared your time and professional expertise to help us improve this paper.

Sincerely,

Shuoye Yang,

College of Bioengineering, Henan University of Technology, Zhengzhou 450001, P. R. China,

  1. The chemical reaction and conjugation process for the synthesis of MSNs-SS@DOX and HT-LMSNs-SS@DOX could be elaborated further for better clarity.

Response: Thanks for the reviewer’s suggestion. The reaction process for the preparation of MSNs-SS@DOX and HT-LMSNs-SS@DOX was rewritten and elaborated.

  1. What is the reasons behind the higher drug loading and encapsulation efficiency observed for Liposomes@DOX compared to MSNs-SS@DOX nanoparticles.

Response: Thanks for the reviewer’s question. The reasons for the higher EE(%) and LE(μg/mg) of Liposomes@DOX compared to MSNs-SS@DOX were as follows: The particle size and loading space of liposomes was larger than MSNs-SS. Besides, the lipid bilayer of liposomes could hinder drug leakage during storage more effectively compared to MSNs-SS carriers.

  1. Please provide the concentrations used for the free drug and drug-loaded MSNs in the cell viability, apoptosis, cell uptake and in cell cycle arrest studies mentioned in the results section.

Response: Thanks for the reviewer’s reminder and suggestion. The concentrations of samples used in the regarding Section were added in the revised manuscript.

  1. While the IC50 values for LMSNs-SS@DOX are significantly different from MSNs@DOX, the same is not observed for LMSNs-SS@DOX and T-LMSNs-SS@DOX or HT-LMSNs-SS@DOX Vs. Please comment for this observation.

Response: Thanks for the reviewer’s question. Indeed, the IC50 values of T-LMSNs-SS@DOX and HT-LMSNs-SS@DOX were notably decreased compared to LMSNs-SS@DOX (especially towards MCF-7/ADR cells), verifying the effect of dual-modification by TPGS and HA. Besides, the lower IC50 value of LMSNs-SS@DOX than that of MSNs@DOX suggested the advantage of hybrid LMSN carriers over MSN carriers with a single structure.

  1. Please include statistical analysis for the IC50 data.

Response: Thanks for the reviewer’s suggestion. The statistical analysis results for the IC50 data were added in the revised manuscript.

  1. Since doxorubicin exhibits autofluorescence, it is surprising that the drug's fluorescence is not visible in the fluorescence microscopy imaging. Please provide your thoughts on this observation.

Response: Thanks for the reviewer’s question. Doxorubicin really had autofluorescence, its fluorescence was not visible only due to the poor quality of previous images. The fluorescence images were redrawn and shown as separate figures, the red fluorescence of DOX could be seen in the new figures.

  1. At the 16-hour time point of HTMSNs for cell uptake, most cells appear viable, yet you mentioned that the HTMSNs is significantly effective in killing more number of cancer cells at 24 hours treatment compared to free drug. Please clarify.

Response: Thanks for the reviewer’s question. In the cellular uptake result (fluorescence microscopy images), blue fluorescence represented the nucleus stained by a nuclear staining agent. Since some dead cells could also exhibit this fluorescence, the fluorescence observation results could not denote cellular survival status totally. Indeed, this experiment was performed to investigate the uptake of nano-formulations. HT-LMSNs-SS@DOX showed a higher uptake efficiency, suggesting that HT-LMSNs-SS@DOX could enter into cells more efficiently than other formulations. Thus, they induced more cells to death at 24 h.

  1. The apoptosis data shows that the ratio of apoptotic cells for MSNs and L-MSNs is not significantly different. However, the IC50 values indicate a significant difference in cytotoxicity between these two nanoparticle formulations when treated on cells. This seeming contradiction between the apoptosis assay and IC50 data for MSNs and L-MSNs needs to be addressed.

Response: Thanks for the reviewer’s question. Compared to the MSNs-SS@DOX group, the cell apoptosis ratios of LMSNs-SS@DOX treatment did not increase notably. However, towards MCF-7/ADR cells, the apoptosis ratios of functionalized LMSNs formulation (T-LMSNs-SS@DOX and HT-LMSNs-SS@DOX) treatments were significantly higher than MSNs-SS@DOX group (especially the HT-LMSNs-SS@DOX). This result suggested that HT-LMSNs-SS@DOX had the potent capacity to induce more apoptosis. Moreover, the IC50 results verified the killing efficacy of nano-formulations against tumor cells. Nevertheless, they induced tumor cells to death through various pathways not only apoptosis. The regarding discussion was added in the revised manuscript.